# The Influence of Temperature on Frequency Modulation Spectroscopy in Atom Gravimeter

**DOI:** 10.3390/s22249935

**Published:** 2022-12-16

**Authors:** Kanxing Weng, Bin Wu, Feichen Wang, Xiaohui Zhang, Yin Zhou, Bing Cheng, Qiang Lin

**Affiliations:** Zhejiang Provincial Key Laboratory of Quantum Precision Measurement, Institute of Optics, College of Science, Zhejiang University of Technology, Hangzhou 310023, China

**Keywords:** gravity measurement, frequency modulation spectroscopy, frequency locking, quantum sensors

## Abstract

Atom gravimeters use locked lasers to manipulate atoms to achieve high-precision gravity measurements. Frequency modulation spectroscopy (FMS) is an accurate method of optical heterodyne spectroscopy, capable of the sensitive and rapid frequency locking of the laser. Because of the effective absorption coefficient, Doppler broadening and susceptibility depend on temperature, and the signal-to-noise ratio (SNR) of the spectroscopy could be affected by temperature. We present a detailed study of the influence of the temperature on FMS in atom gravimeters, and the experimental results show that the SNR of the spectroscopy is dependent on temperature. In this paper, the frequency of the reference laser is locked by tracking the set point of the fringe slope of FMS. The influence of the frequency-locking noise of the reference laser on the sensitivity of the atom gravimeter is investigated by changing the temperature of the Rb cell without extra operations. The method presented here could be useful for improving the sensitivity of quantum sensors that require laser spectroscopic techniques.

## 1. Introduction

Since the first quantum sensor based on atom interferometry was developed in 1990s [1,2,3], atom interferometry techniques have been demonstrated to be extremely sensitive and accurate methods. These sensors enable precise measurements in lots of fields, such as the determination of fundamental constants [4,5,6,7,8,9,10,11], measurement of the rotation rate [12,13,14,15,16], gravitational accelerations [17,18,19,20], gravity-field curvature [21], and gravity-field gradient [22,23,24,25,26,27]. At present, the atom gravimeter has demonstrated better performance than its classical counterparts [28,29,30]. Laser spectroscopic techniques are vital methods of experiments used to study atom properties [31,32], including saturated absorption spectroscopy (SAS) [33,34], modulated transfer spectroscopy (MTS) [35,36], frequency modulation spectroscopy (FMS) [37,38], etc. Indeed, both FMS and MTS have the advantage of not relying on the polarization of lasers and fluctuations in laser intensity. However, the FMS laser system is simpler and has a good signal-to-noise ratio, as well as a better short-term stability performance [39], which is conducive to improving the short-term sensitivity of the atomic gravimeter and to the integration of the miniaturized atom-gravimeter laser system. Among all these methods, FMS is a typical method of optical heterodyne spectroscopy capable of sensitive and rapid measurements of the absorption and dispersion associated with weak, narrow spectral features.

FMS is based on converting the imposed frequency modulation into an amplitude modulation as an atom transition is encountered [40]. FMS can be classified into regimes depending on the modulation depth *M* or frequency *ω*. When the modulation depth (M ≪ 1) is small and the modulation frequency (ω > Г, where the Г is the natural line width) is large, it is called frequency modulation (FM). On the contrary, when the modulation depth is large (M ≫ 1 and ω < Г) and the modulation frequency (ω < 1 MHz) is smaller, it is named wavelength modulation (WM) spectroscopy [41]. Based on the method of frequency modulation, FMS can effectively circumvent the problem of laser intensity noise and offer a method for phase-sensitive detection with high sensitivity at the per billion level [42]. Therefore, FMS can be used as an ultra-high precision-frequency reference in the atom sensors. In addition, the transition probability population of the atom gravimeter, which is measured by collecting fluorescence signals of atoms in each of the two hyperfine states. As such, the frequency noises directly affect the signal-to-noise ratio of fluorescence signals and ultimately jeopardize measurement accuracy [19].

In this paper, a unique experimental method has been applied to characterize the influence of temperature on the signals of FMS and thus on the performance of an atom gravimeter by observing frequency noise and sensitivity. We firstly describe the experimental setup, and then analyze the influence of temperature on FMS. In particular, we investigate the relationship between the temperature and the detection noise of the atom gravimeter; thereby, the explanation of its contribution to the sensitivity of the atom gravimeter is discussed. Finally, the results show that the short-term sensitivity of the atom gravimeter increased from $1305\;\mu \mathrm{Gal}/\sqrt{\mathrm{Hz}}\;$ to $\;702\;\mu \mathrm{Gal}\;/\sqrt{\mathrm{Hz}}$ via an optimization of the temperature for FMS. This research yields an efficient method for the improvement of the sensitivity of atomic sensors.

## 2. Experimental Setup

The laser system of our atom gravimeter has been described in detail elsewhere [43,44]. As shown in Figure 1, the laser system consists of two external-cavity diode lasers (ECDLs) at 780 nm, and the ECDL1 is called the reference laser. The frequency of the reference laser is locked to the *F* = 1 → *F*′ = 1 transition of the D_2_ line of ^87^Rb via FMS. The theory of FMS has been described in detail in [38,40]. It utilizes an external phase modulator to produce the WM and is capable of detecting weak absorption or dispersion features with the full spectral resolution characteristic of lasers. The modulator converts the single-axial-mode laser input into pure FMS with a low modulation index. In addition, both the absorption and the dispersion associated with the spectral feature can be separately measured by monitoring the phase and amplitude of the RF heterodyne beat signal that occurs when FMS is distorted by the effects of the spectral feature on the probing sideband. Figure 1 describes a schematic of the typical experimental arrangement for FMS, where the driving frequency ω of the electro-optic modulator (EOM) is 12.5 MHz. The digital locking module refers to the FPGA-based laser frequency locking with an auto-locking module, and it can auto-lock the frequency of the ECDL1 by tracking the set point of the fringe slope of FMS. Additionally, a heating wire is implemented around the cell to control the temperature of the Rb cell. The frequency of the ECDL2 is controlled by beating with the reference laser. Raman light is generated by the AOM frequency shift after the ECDL1 and EDCL2 are combined. In the cooling beam and the detection beam, the laser of ECDL1 is used as the repumping laser, and the light of the ECDL2 can quickly switch between cooling and detection via frequency control.

In the experiment performed here, the laser beams are transmitted to the vacuum system via a polarization-maintaining fiber. The cooling beam is combined with a gradient magnetic field to form a magneto-optical trap for trapping and cooling 10^8^ atoms in 280 ms, before cooling them down to 5 μk in 20 ms. Then, the atoms are selected in *F* = 1 using a microwave pulse. During their free fall, atom interferometry is performed using a sequence of three counter-propagative Raman laser pulses (π/2-π-π/2), separated by free evolution times *T* = 55 ms. Finally, the whole sequence ends at the bottom of the experiment with the successive time-of-flight (TOF) fluorescence detection of the populations of the two interferometer output ports, thanks to the state-labeling method.

## 3. Theory

The demodulated dispersive signals of FMS are shown in Figure 2. The blue curve in Figure 2a illustrates a partially demodulated dispersive signal of FMS obtained by a digital laser lock module where the dispersive signals are the ^87^Rb D_2_ line *F =* 1 → *F*′ *=* 0, *F*′ *=* 1, *F*′ *=* 2 formants. The frequency of the ECDL1 is locked in *F =* 1 → *F*′ *=* 1 by tracking the set point of the fringe slope. The red line in Figure 2b is a linear fitted curve to the side of the *F =* 1 → *F*′ *=* 1 fringe, and the slope *γ* (${\gamma =\rho }_{\mathrm{vpp}}/\omega$, where $\rho_{\mathrm{vpp}}$ is the peak-to-peak value of the amplitude *ρ*, *ω* is the modulation frequency) is maintained via the digital locking module.

The frequency-locking noise δν is defined by δν=δρ/γ. Therefore, the frequency-locking noise δν depends on the slope *γ*, and the higher the value of γ, the lower the frequency-locking noise δν.

In the following, we analyze the relationship between the temperature T and slope *γ* of the demodulated dispersive signal of FMS. Based on the theory of FMS [40], the modulated laser ECDL1 with intensity *I*_0_ travels through the Rb cell with the absorption coefficient *α* and length *l* = 150 mm. Then, the intensity of the incident modulated laser *I*_0_ is reduced to *I_s_* as:(1)$$I_{s}=I_{0}\exp(-2\delta )\times (-\delta_{1,-1}\times M),$$
where *I*_s_ is the detection light intensity of FMS, *δ* is the amplitude attenuation  (δ=αl/2), Δ*δ* is the difference in first-order sideband signal absorption $\delta_{1,-1}=\delta_{1}-\delta_{-1}$, and *M* is the modulation depth.

The absorption coefficient *α*, as a function of the laser angular frequency *ω* near an atom transition with resonant angular frequency *ω_P_*, is given approximately as the simple sum of the Doppler broadenings *α_D_* due to the random thermal motion of the atoms and the Lorentzian functions in the Doppler limit [45]:(2)α=∑P|μP|2exp[−(ω−ωP)2/(ku)2]+kImχ≡αD+kImχ,
where *k* is the wave number of the laser, *ꭓ* is the susceptibility, and ${\left|\mu_{P}\right|}^{2}$ is the hyperfine transition probability. The most probable velocity of *u* is given by $u={\left({2\;k}_{B}T/m\right)}^{1/2}$, where *T* is the temperature of the Rb cell, *k_B_* is Boltzmann’s constant, and *m* is the atom’s mass.

Here, let us consider the effect of the temperature on the absorption coefficient *α*. The absorption coefficient *α* needs to be modified, and the modified absorption coefficient can be represented by *α^corr^*:(3)$$\alpha^{{}_{corr}}\propto n(T)S\sigma (\omega )\alpha ,$$
where the number density of the Rb cell of *n*(*T*) is given by $n(T)=P_{V}{/(k}_{B}T)$, *P_V_* is the pressure of saturated vapor, *S* is the hole-burning area, and *σ*(*ω*) is the cross-section. Because of the effective atomic number density, the Doppler background absorption coefficient and the electrodeposition rate are affected by the temperature. As shown in Figure 3, the relationship between temperature *T* and spectral absorption intensity *β* can be given by Equation (4). For temperatures below 59 °C, with an increase in the temperature of the Rb cell, the relative intensity *β* trends with two decaying exponents with a maximum at *T* = 44 °C. The optimum temperature *T* = 44 °C is determined by the Rb cell length *l*.
(4)$$\beta =\frac{I_{s}}{I_{0}}\propto \exp(-\alpha^{corr}l)-\exp(-\alpha_{D}{}^{corr}l),$$
where *β* is the relative intensity of the absorption without the Doppler background, *α_D_^corr^* is the correctional Doppler broadenings, and $\alpha_{D}{}^{corr}\propto n(T)S\sigma (\omega )\alpha_{D}$.

The relationship between the $\rho_{\mathrm{vpp}}$ of demodulated dispersive signal and the relative intensity of absorption *β* can be expressed as $\rho_{\mathrm{vpp}}=\gamma \omega \propto \beta$.

## 4. Experimental Results

According to the analysis above, we can intentionally modify the temperature of the Rb cell to change the slope *γ* of FMS to investigate the effect of the frequency-locking noise δν on the sensitivity of the atom gravimeter. The FMS signals obtained by changing the temperature of the Rb cell are illustrated in Figure 4a. The amplitude of the FMS signal is affected by various factors with respect to the applied temperatures. For simplicity, we shall only discuss the transition of the ^87^Rb D_2_ line *F* = 1 → *F*′ = 1. As shown in the partial enlargement in Figure 4a, we can observe the relationship between the temperature *T* ∈ (34~59 °C) and the amplitude *ρ*. For temperatures *T* below 44 °C, the amplitude *ρ* increases with the *T*, and the amplitude *ρ* decreases with *T* as the result of the influence of the effective atom number density, Doppler broadening, and susceptibility. The results of this experiment validate the previous theoretical analysis $\rho_{\mathrm{vpp}}\propto \beta$. Therefore, by heating the Rb cell, the amplitude *ρ* of FMS (the ^87^Rb D_2_ line *F* = 1 → *F*′ = 1) can be adjusted efficiently without introducing additional noise.

Figure 4b clearly displays the slope *γ* of the demodulated dispersive signal with varying temperatures, and we find good agreement of the slope *γ* with the relative intensity of absorption *β*, and the slope *γ* reaches a peak *γ* = −0.073 V/MHz at *T* = 44 °C. Therefore, we determine that the optimum slope is *γ* = −0.073 V/MHz due to the definition of frequency-locking noise δν=δρ/γ. In addition, the slope *γ* of FMS could also be modified by adjusting the normalized frequency *R* [40]. *R* is defined by R=(ω−Ω)/(ΔΩ/2), where Ω is the line center frequency and ΔΩ is the full width at half maximum. However, this method may introduce other systematic noises, such as electronic noises from the frequency modulator.

The principle of the atom gravimeter is based on the coherent manipulation of atom wave packets by the two-photon-stimulated Raman transitions. The transition probability *P* is given by: P=1/2(1−cosϕ), where *ϕ* is the phase shift accumulated by the atom matter waves along two interferometer arms. For an ideal atom gravimeter, $\phi =(\alpha_{0}-{\vec{k}}_{\mathrm{eff}}\times \vec{g})T^{2}$, where *α*_0_ is the frequency chirp of the Raman lasers applied to compensate the gravity-induced Doppler shift, $\vec{g}$ is the local gravitation acceleration and *T* is the time separation between pulses. In addition, in the presence of a uniform gravity field, the phase shift is denoted by $\phi =k_{\mathrm{eff}}gT^{2}$. However, the transition probability *P* is deduced from the population in *F* = 1 and *F* = 2, and the detection laser is used to obtain the population of ^87^Rb in each of the two hyperfine states by measuring the resonant fluorescence signal. As such, the transition probability *P* depends on the frequency-locking noise δv of the reference laser.

Therefore, we assess the detection noise $\sigma_{D}$ independently of any interferometer by preparing a superposition using a π/2 microwave pulse. Then, the atoms are distributed on the ground state *F* = 1 and *F* = 2, as shown in Figure 5.

The result of this measurement is shown Figure 6a, and the detection noise can be defined as $\sigma_{D}=2\sqrt{2}\sigma_{P}/CkT^{2}$ ($\sigma_{P}$ is the standard deviation of transition probability and *C* = 0.3 is the contrast of the atom interferometer). In order to highlight the effect of the frequency-locking noise *δν* on the sensitivity of the atom gravimeter, we deliberately selected a set of slopes *γ* with a large difference of *γ* = −0.073 V/MHz, −0.044 V/MHz, −0.028 V/MHz for the experiments. Figure 6b shows the short-term sensitivity of $702\;\mu \mathrm{Gal}/\sqrt{\mathrm{Hz}}$ (0.073 V/MHz), $1077\;\mu \mathrm{Gal}/\sqrt{\mathrm{Hz}}$ (−0.044 V/MHz), and $1305\;\mu \mathrm{Gal}/\sqrt{\mathrm{Hz}}$ (−0.028 V/MHz), respectively. In addition, the stability of the gravity measurement improves as *τ*^−1/2^ reaches 22 μGal, 34 μGal, and 40 μGal after 1000 s. The results have demonstrated the effect of the frequency-locking noise on the long-term or short-term sensitivity of the atom gravimeter. The lower the frequency-locking noise *δν*, the better the sensitivity of the atom gravimeter Δ*g*.

In summary, we have introduced a frequency-locking scheme based on FMS technology, and then analyzed the effect of Rb cell temperature on the frequency-locking noise. By analyzing the fluctuation of the transition probability *σ_P_*, the relationship between the frequency-locking slope *γ* and the *σ_P_* is obtained, and the relationship between the sensitivity of the atom gravimeter and the frequency-locking noise is carried out. The results show that the short-term sensitivity of the atom gravimeter increased from $1305\;\mu \mathrm{Gal}/\sqrt{\mathrm{Hz}}$ to $702\;\mu \mathrm{Gal}/\sqrt{\mathrm{Hz}}$ by optimizing the slope *γ*.

## 5. Discussion

We have investigated the effect of frequency-locking noise on the sensitivity of the atom gravimeter by modifying the slope *γ* of FMS. Furthermore, the slope *γ* of FMS was only modified by setting different temperatures in the Rb cell, which can effectively avoid other uncertain noises. Through monitoring the fluctuation of the transition probability *σ_P_*, we found that the frequency-locking noise directly affects the detection noise, and that setting the appropriate Rb temperature (*T* = 44 °C) can effectively suppress this noise. As the experimental results show, the short-term sensitivity of the atom gravimeter increased from $1305\;\mu \mathrm{Gal}/\sqrt{\mathrm{Hz}}$ to $\;702\;\mu \mathrm{Gal}/\sqrt{\mathrm{Hz}}$, while the long-term sensitivity of the atom gravimeter increased from 40 μGal to 22 μGal after the integration of 1000 s. After optimizing the laser frequency noise, the atom gravimeter sensitivity was mainly limited by vibration noise and laser phase noise.

## Figures and Tables

**Figure 1 sensors-22-09935-f001:**
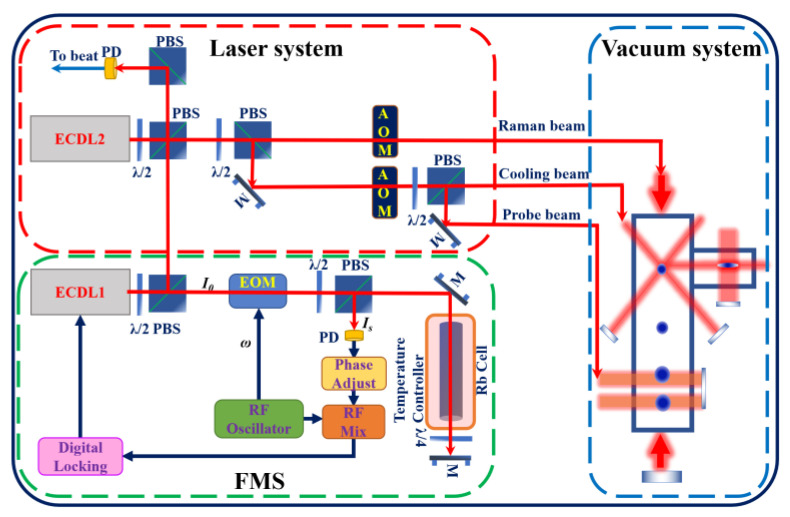
Scheme of frequency modulation spectroscopy (FMS). ECDL is the external-cavity diode laser, EOM is the electro-optic modulator, PBS is the polarizing beam splitter, PD is the photo diode, λ/4 is the quarter-wave plate, M is the mirror.

**Figure 2 sensors-22-09935-f002:**
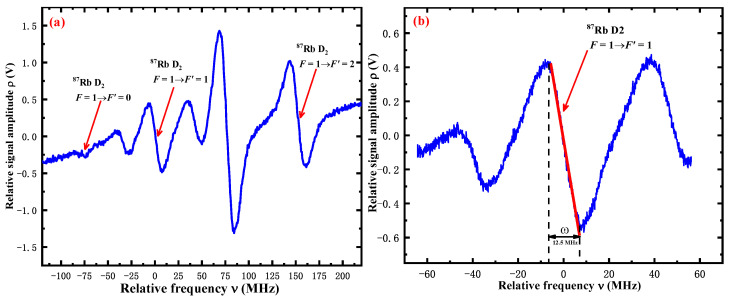
The results of the demodulated dispersive signal of FMS. (**a**): The measurement of the dispersive curve of the ^87^Rb D2 line *F* = 1 → *F*′ = 0, *F*′ = 1, *F*′ = 2. (**b**): The measurement of the dispersive curve of the ^87^Rb D2 line *F* = 1 → *F*′ = 1, and the red line with the slope *γ* = −0.07959 V/MHz is the linear fitting of the side of the *F* = 1 → *F*′ = 1, and the modulation frequency *ω* = 12.5 MHz.

**Figure 3 sensors-22-09935-f003:**
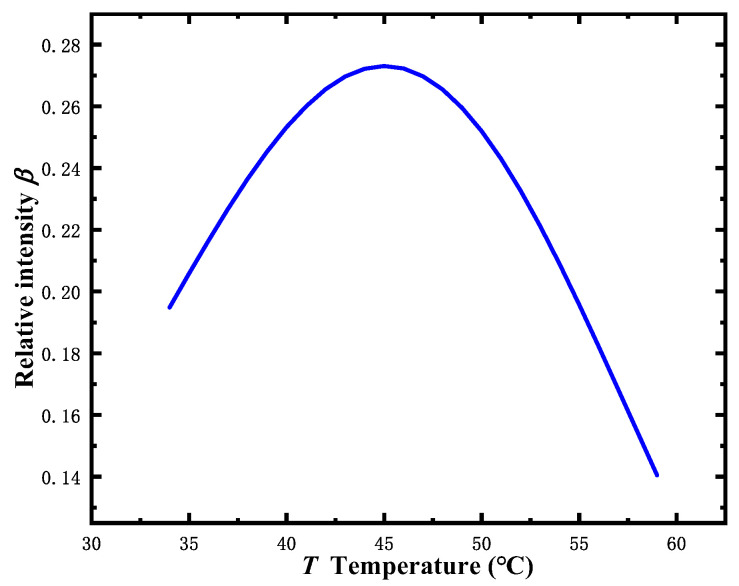
The theoretical calculation of the relationship between *β* and *T*.

**Figure 4 sensors-22-09935-f004:**
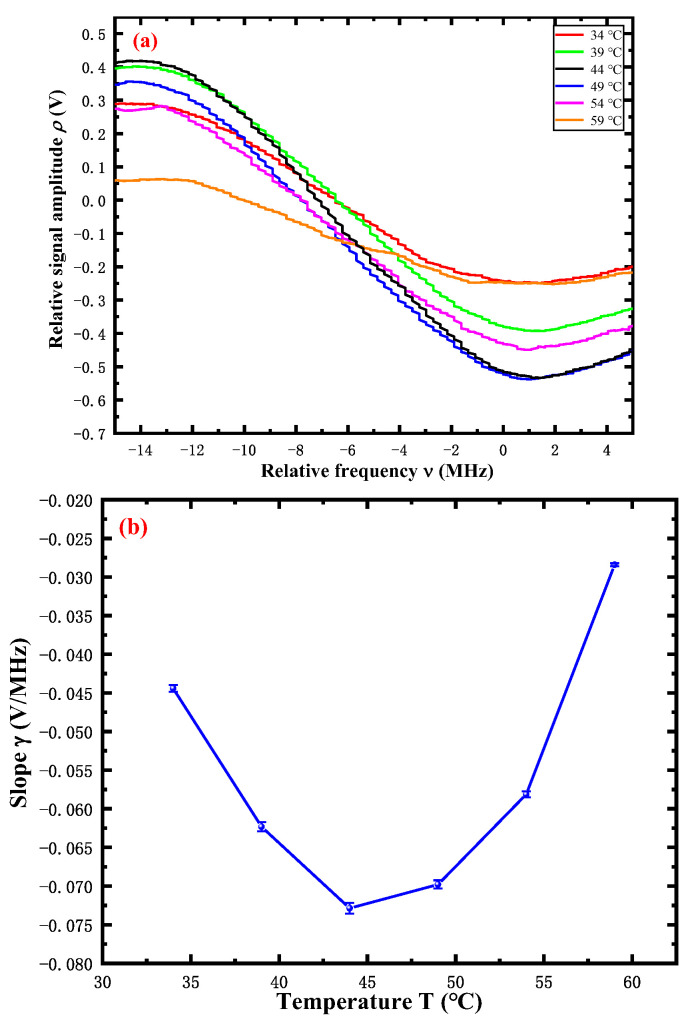
The results of the demodulated dispersive signal and its slope versus temperature of the Rb cell. (**a**): The measurements of the demodulated dispersive signal with varying temperatures *T* ∈ [34 °C, 59 °C]. (**b**): The blue points represent the slope of the demodulated dispersive signal fringe of the ^87^Rb D2 *F* = 1 → *F*′ = 1.

**Figure 5 sensors-22-09935-f005:**
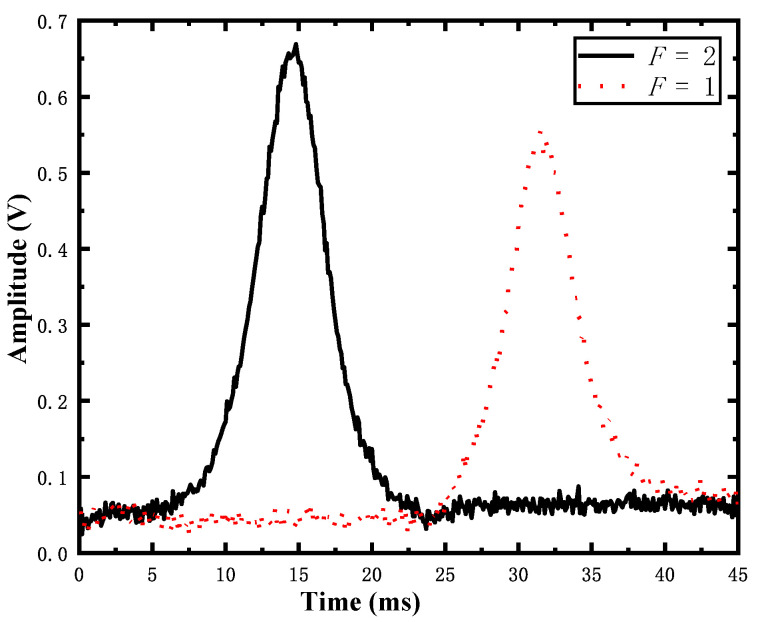
Typical plot of detection signals after π/2 microwave pulse; the two curves are for the *F* = 1 and *F* = 2 channels, respectively.

**Figure 6 sensors-22-09935-f006:**
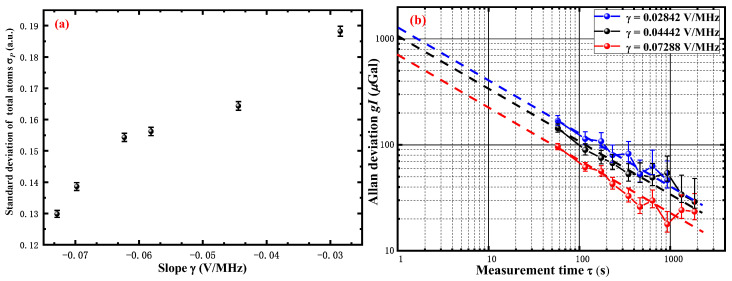
The Allan deviation of the gravimeter measurements and the fluctuation of the transition probability *σ_P_*. (**a**) The standard deviation of detection *σ_P_* with the slope *γ*. (**b**) The Allan deviation of the gravimeter measurement with the various slope *γ*. The dashed lines illustrate the *τ*^−1/2^ scaling, which corresponds to the white noise.

## Data Availability

Not applicable.

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
