# Peer review of "The Influence of Temperature on Frequency Modulation Spectroscopy in Atom Gravimeter"

_sensors, 2022, doi:10.3390/s22249935_

Round 1
Reviewer 1 Report
In the manuscript, the authors present their study on the influence of the temperature of FMS in atom gravimeter. They modulate the temperature and find the optimal peak-to-peak value of the signal amplitude as well as the slop of the demodulated dispersive signal. The authors also assess the effect of the frequency locking noise on the sensitivity of atom gravimeter through assessing the detection noise. Finally, they conclude that the short-term sensitivity could be improved by 2 folds.
There is no particularly new or novel science or technology reported here as the effect of temperature on FMS has already been studied in many papers, I guess the authors acknowledge this point and they highlight the effect of the frequency locking noise on the sensitivity of the atom gravimeter. A work which investigates the frequency noise in an atom gravimeter is a valuable contribution and is appropriate for a publication in the Sensors. However, despite the subject of the paper being appropriate for a publication, I believe that not enough elements have been given. I therefore recommend that the paper be resubmitted after the authors have addressed all my criticisms.
The detail of my criticisms of their work which lead to my decision are present in the following remarks:
1. In the first paragraph of Introduction, more information should be offered about the merits of FMS compared to the other laser spectroscopic techniques which lead to their choice of using FMS. As far as I know, most of the groups engaged in atom gravimeter in the world take MTS.
2. Although FMS has been described in detail in the cited references [36,37], I believe a brief description is yet necessary in proper position in the manuscript.
3. In Fig.1, two counter-propagating lasers are used in Rb cell to remove Doppler broadening. However, the amplitude of the spectroscopy is also related to the power ratio of the counter-propagating lasers. When changing the temperature of Rb cell, the power ratio will change due to the variation of α, there is no any discussion about the effect of light power ratio.
4. There should be an explanation why there is an inflection point when T=44℃. The author should elucidate the physical mechanism, or at least list the parameters of their calculation.
5. Frequency noise deteriorates the sensitivity of atom gravimeter through detection, is there any other way?
6. The contribution of frequency noise on sensitivity is calculated from the transition probability after a π/2 microwave pulse considering the contrast is 1, but the contrast can not reach 1 in actual experiments. Thus the sensitivity is overestimated.
Besides, why not study its effect in real gravity measurement sequence, namely in a π/2-π-π/2 Raman sequence? In addition, the authors should do a comprehensive study of the effect of the frequency noise to atom gravimeter.
7. δ in Line 100 should be replaced by other character to distinguish from the meaning of δ in Line 86.
8. Reference [28] is the same with [16].
9. The unit of short-term sensitivity should be converted to the unit of gravitational acceleration (considering their pulse separation) to give readers a more intuitive impression since this manuscript discuss the influence of temperature on FMS in atom gravimeter.
10. There are many minor format problems, such as superscript and subscript, uppercase and lowercase, indent, singular and plural and typos. They distribute in (BUT NOT LIMITED IN) the following lines: 72, 91, 96, 98, 104, 108, 109, 113, 114, 116, 120, 201, Fig 6…
Authors should check their manuscript more carefully.
Reviewer 2 Report
Please see my full report attached.

Round 2
Reviewer 1 Report
From the reponse and the revised manuscript, most of the comments in my previous report have been addressed. I believe this paper is now suitable for a publication in Sensors and recommend its publication.
A few minor suggestions:
- ‘a parabola dependence‘ in abstract is in contradiction with Eq. (4)
- In line 77, 'rf' →RF
- In line 91, ‘108’,8 is superscript
- In line 131, 'αD', D is subscript
- In line 149, 'and' is unnecessary
- In line 208 or 209, the value of C would better be presented
- In Figure 1, the font size is too small to easily read
